# Drivers of soil microbial and detritivore activity across global grasslands

Julia Siebert[1,2,28], Marie Sünnemann [1,2,28✉], Yann Hautier [3], Anita C. Risch[4], Jonathan D. Bakker[5], Lori Biederman[6], Dana M. Blumenthal[7], Elizabeth T. Borer[8], Miguel N. Bugalho[9], Arthur A. D. Broadbent [10], Maria C. Caldeira [11], Elsa Cleland[12], Kendi F. Davies[13], Anu Eskelinen[2,14,15], Nicole Hagenah[16], Johannes M. H. Knops[17], Andrew S. MacDougall[18], Rebecca L. McCulley [19], Joslin L. Moore[20,21], Sally A. Power [22], Jodi N. Price[23], Eric W. Seabloom[8], Rachel Standish [24,25], Carly J. Stevens [26], Stephan Zimmermann[27] & Nico Eisenhauer [1,2]

Covering approximately 40% of land surfaces, grasslands provide critical ecosystem services that rely on soil organisms. However, the global determinants of soil biodiversity and functioning remain underexplored. In this study, we investigate the drivers of soil microbial and detritivore activity in grasslands across a wide range of climatic conditions on five continents. We apply standardized treatments of nutrient addition and herbivore reduction, allowing us to disentangle the regional and local drivers of soil organism activity. We use structural equation modeling to assess the direct and indirect effects of local and regional drivers on soil biological activities. Microbial and detritivore activities are positively correlated across global grasslands. These correlations are shaped more by global climatic factors than by local treatments, with annual precipitation and soil water content explaining the majority of the variation. Nutrient addition tends to reduce microbial activity by enhancing plant growth, while herbivore reduction typically increases microbial and detritivore activity through increased soil moisture. Our findings emphasize soil moisture as a key driver of soil biological activity, highlighting the potential impacts of climate change, altered grazing pressure, and eutrophication on nutrient cycling and decomposition within grassland ecosystems.

A full list of author affiliations appears at the end of the paper.

Grassland systems covering approximately 40% of the world's terrestrial surface, encompass a wide variety of habitats for soil organisms[1,2], which play key roles in delivering ecosystem functions such as nutrient cycling and decomposition[3–6]. In this context, the key players are soil microorganisms and detritivores such as earthworms, isopods, millipedes, and enchytraeids, which primarily feed on litter and organic materials. Their collective efforts break down organic matter, thus supplying vital nutrients to plants[7]. Soil organism activity is strongly driven by temperature, soil moisture[8–11], and global change factors, including increased nutrient inputs and alterations in the range, abundance, and distribution of aboveground herbivores. However, we lack a broad understanding of how nutrient inputs and herbivory influence soil communities and ecosystem functions in grasslands. At the same time, such soil organisms may be important mediators of ecosystem responses to global change[2,12,13]. Further, a lack of spatially replicated studies means that we cannot predict how plant productivity, grazing, or local abiotic characteristics may mediate nutrient and herbivory effects on soil organisms[14].

Herbivores can play a crucial role in shaping grasslands by facilitating diverse plant communities and maintaining ecosystem functioning[14]. For example, wild herbivores may selectively consume abundant plant species, altering species composition[15–17] and can contribute to maintaining plant diversity by reducing competition for light[18]. Moreover, herbivores impact nutrient cycling in grasslands by consuming live plant material and modifying the quantity and quality of organic inputs to the soil, e.g. via excreta, and via changes in soil abiotic conditions[14,19,20]. At the same time, large native herbivore densities may be reduced via hunting or land conversion, and in many cases, they are replaced by large numbers of domestic livestock[21,22]. Soil communities, processes, and structure are strongly affected by wild and domestic herbivores, with important consequences for soil biological activity and ecosystem multifunctionality[14,23–25]. Herbivores may enhance soil biological activity by depositing easily-degradable dung and urine or, particularly under fertile conditions, inducing compensatory growth[19,26–28]. In contrast, in relatively unproductive systems, grazers may preferentially feed on the few available nutrient-rich plants, on which many soil organisms also depend, resulting in poorer quality of litter, which reduces biological activity[15,19,29,30]. Additionally, aboveground herbivores may create harsh environmental conditions for soil organisms through soil compaction, negatively affecting pore space and water infiltration as well as increasing the cover of bare soil, resulting in high temperature fluctuations compared to vegetated areas[31,32]. At the same time, the interaction between herbivory and nutrients can be context-specific, as it may vary based on the specific plant species and local site conditions[33].

Predictions suggest that the disruption of the nitrogen cycle could cause nitrogen (N) deposition to double in the future[34,35]. The same applies to phosphorus (P) inputs, which have globally increased compared to preindustrial levels[36,37]. The growth and biomass production of plants depend on nutrients such as nitrogen, phosphorous, and potassium, and most grasslands are limited in productivity by nutrient inputs[38,39]. Nitrogen inputs may increase the activity of soil organisms by increasing the amount and quality of plant material that enters the soil system[10,20], but have also been shown to reduce detritivore activity[40]. The same applies to soil microbes, as long-term nitrogen inputs have been shown to have negative effects[41]. While the effects of phosphorous inputs on microbial activity remain less understood, it is known that phosphorous limitation can impede decomposition[41–43]. Globally, nitrogen to phosphorous ratios are increasing, leading to a prevalence of phosphorous

limitation in soils[36]. This limitation can further inhibit microbial activity, which in turn can impact biological decomposition processes[44]. Additionally, although soil microbes are generally less susceptible to potassium deficiency than plants[45], they still benefit from increased nutrient inputs, including potassium and micronutrients from plants that have sufficient nutrient supply. Given these context-dependent effects of nutrient addition and herbivory on soil processes, we need standardized manipulations of herbivores and nutrients across experimental and environmental gradients.

To improve our understanding of how fertilization and herbivory may alter ecosystem functioning belowground, we investigated the effects of nutrient enrichment (NPK fertilization) and herbivore reduction on soil microbial and detritivore activity across grasslands worldwide. This globally-coordinated study of soil biota was carried out within the Nutrient Network experiment[46], with sites in North and South America, Europe, Asia, and Australia that represent a wide range of grassland habitats and environmental conditions (Fig. 1a; Table S1). In 2015, we used standardized bait (bait lamina strips) at 18 sites to assess soil detritivore feeding activity[47], and analyzed soil samples from 26 sites for soil microbial activity (microbial respiration)[48]. Throughout the manuscript, we use the term 'biological activity' to encompass both activities. We used structural equation models to test which biotic (plant community properties[49]) and abiotic properties (soil water content[11]) determine soil biological activity worldwide. We hypothesized that (1) reducing aboveground herbivores would result in a decrease in belowground activity rates. Furthermore, we expected (2) the impact of added nutrients on soil biological activity would depend on carbon inputs, with increased plant biomass due to nutrient additions being associated with higher soil biological activity. With the two treatments in combination (3), the positive effect of nutrients on soil activity would be stronger than the negative effect of reduced herbivory, leading to a net increase in soil biological activity. Here, we expected that the positive effect of nutrients on soil activity would be stronger than the negative effect of reduced herbivory.

## Results

**Effects of nutrient addition and herbivore exclusion.** Soil detritivore feeding activity ranged from 0.94% to 77% of available bait substrate removed (Fig. S1a). Soil microbial respiration ranged from 0.27 μl $O_2$ $h^{-1}$ $g^{-1}$ soil dry weight to 8.93 μl $O_2$ $h^{-1}$ $g^{-1}$ soil dry weight. (Fig. S1b). Nutrient addition had no significant effect on detritivore feeding activity ($F = 0.08$; $p = 0.78$) and soil microbial activity ($F = 0.94$; $p = 0.333$). Despite high among-site variation (Fig. S2), herbivore reduction had a positive effect on detritivore feeding (Fig. 1b; $F = 3.60$; $p = 0.06$), resulting in higher activity levels when herbivores were reduced (+16.7%). At the same time, herbivore reduction did not affect soil microbial activity (Fig. 1c; $F = 0.29$; $p = 0.59$) Similarly, there was no interactive effect of NPK fertilization and herbivore reduction on detritivore ($F = 0.21$; $p = 0.65$) and microbial ($F = 0.63$; $p = 0.43$) activity.

**Structural equation model analyses.** The site-specific environmental conditions and treatments also had strong effects on the soil environment and the associated plant community, which became more apparent when the interdependence of variables was considered. Mean annual precipitation (MAP) and soil water content were positively correlated (Fig. S6) and structural equation modeling shows that MAP and soil water were positively associated with soil detritivore and microbial activity (Figs. 2a and 3a). At the same time, soil biological activity rates increased with higher amounts of MAP and soil water content, regardless of other treatment conditions (Figs. S4a, b and S5). Reflecting our

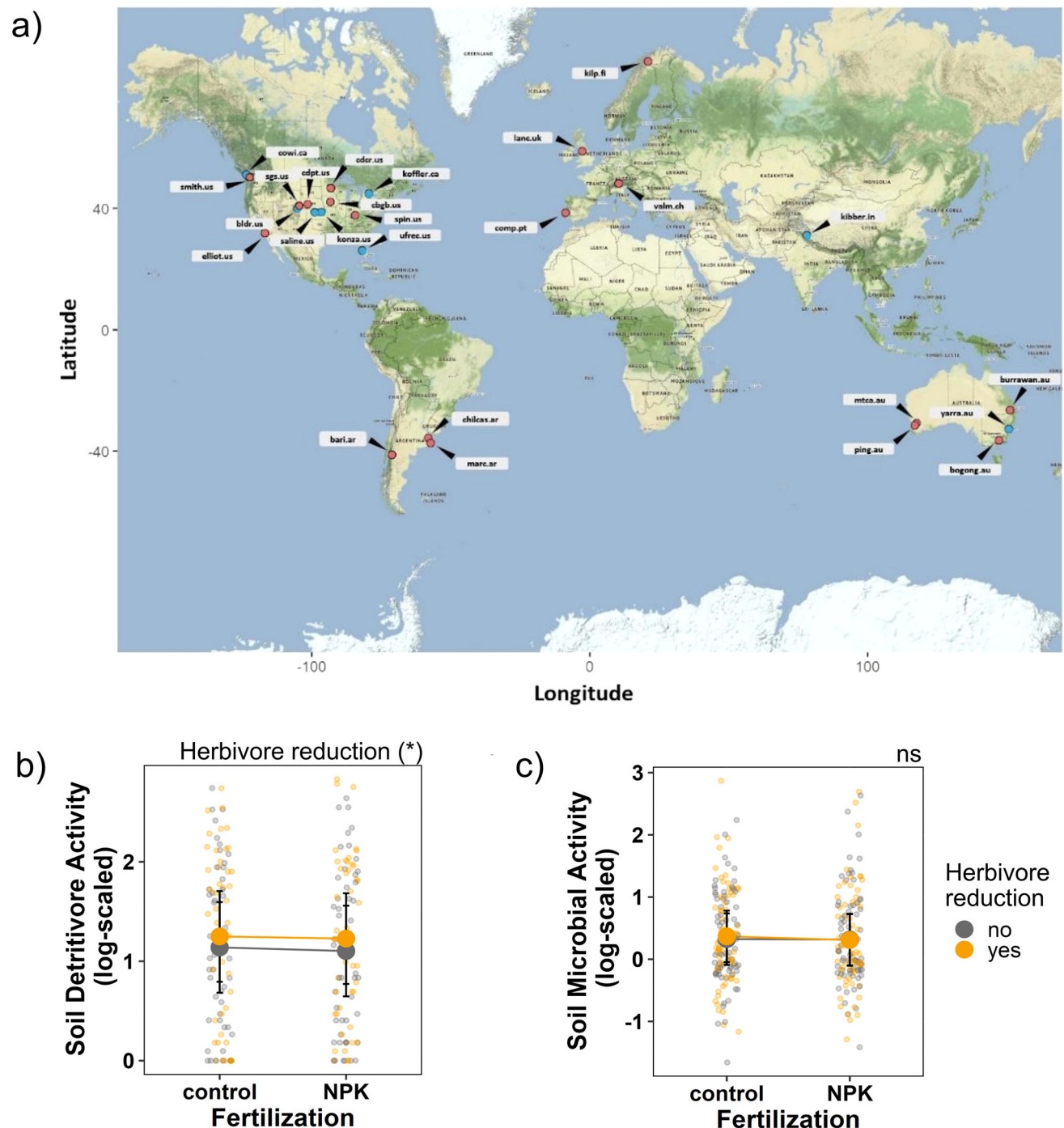

**Fig. 1 Global distribution and treatment effects. a** Global map of all participating sites in the study. Red dot = data on soil microbial and detritivore activity (n = 18 sites); blue dot = data on soil microbial activity only (n = 26 sites). **b, c** Show two figures where we tested the effect of NPK fertilization, herbivore reduction, and the interactive effect of NPK fertilization and herbivore reduction on soil detritivore activity (log-scaled) and soil microbial activity (log-scaled). Points are raw observations; error bars indicate 95% confidence intervals. Significance levels: (*) p-value = 0.06, ns not significant.

results from linear mixed-effects models, nutrient addition had no direct effect on soil biotic activity. However, our SEM model revealed that herbivore reduction directly increased detritivore activity and indirectly increased activity of all soil microbes and detritivores via increasing soil water content (Table S3; Figs. 2a, c and 3a, c). Plant biomass, which increased with site MAP and both NPK and herbivore reduction treatments, was related with lower soil microbial activity (see also Fig. S6). At the same time, we found detritivore and microbial activities to be significantly positively correlated (Fig. 4; F = 9.15; p = 0.003).

## Discussion

We conducted a globally-distributed experiment assessing the responses of soil biological activity to nutrient addition and herbivory at 26 sites, spanning five continents and multiple grasslands. Soil microbial and detritivore activity were associated with similar drivers at the global scale. Soil biological activity increased with MAP and soil moisture, suggesting that future climatic changes related to alterations in the amount and frequency of precipitation as well as evapotranspiration[50] may have major consequences for grassland ecosystem functioning. To

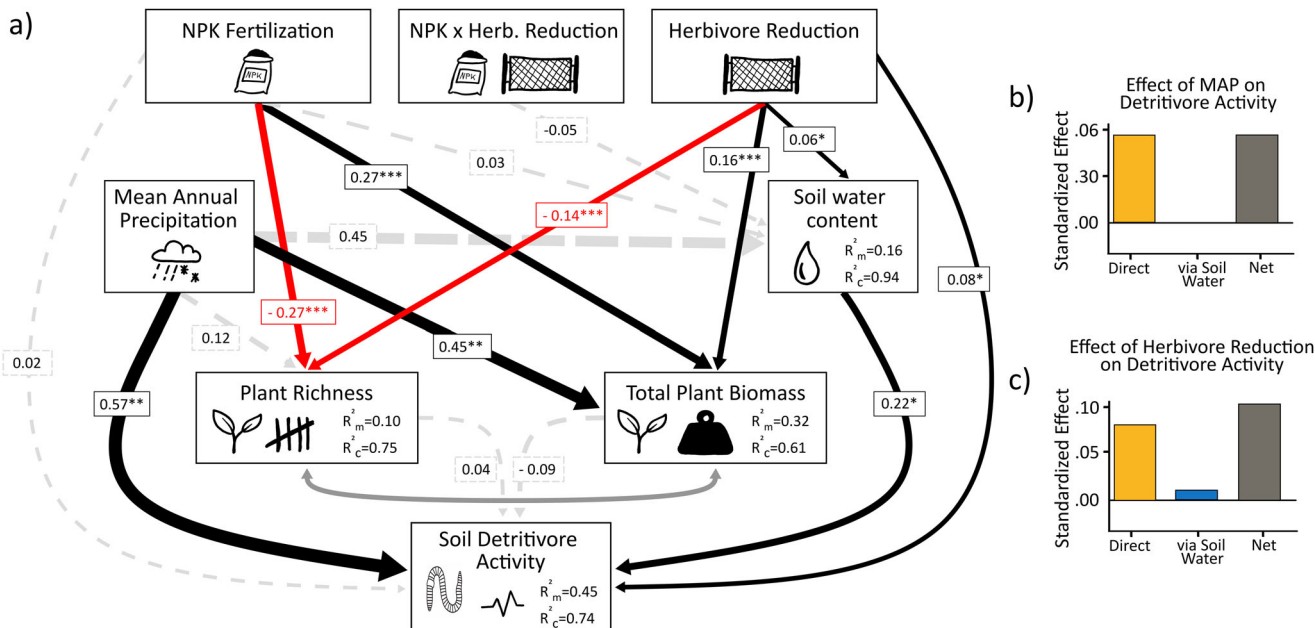

**Fig. 2 Structural equation model of soil detritivore activity. a** Soil detritivore activity as a best-fit Structural Equation Model showing the effects of NPK fertilization and herbivore reduction (Fisher's $C = 1.88$; $P = 0.758$; d.f. = 4; 18 sites). Black arrows indicate significant positive and red arrows indicate significant negative effects in the model ($P < 0.05$). Dashed gray arrows indicate non-significant effects ($P > 0.05$) that remain in the model based on AIC. Dark gray double-headed arrows indicate paths that were treated as correlated errors in the model. Arrow widths are proportional to their effect sizes. Numbers along the arrows are standardized path coefficients. Marginal $R^2_m$: model variation explained by fixed effects; conditional $R^2_c$: model variation explained by both fixed and random effects. Significance levels: *$p < 0.05$; **$p < 0.01$; ***$p < 0.001$. **b** Direct, indirect, and net effect of MAP on soil detritivore activity, and **c** direct, indirect, and net effect herbivore reduction on soil detritivore activity.

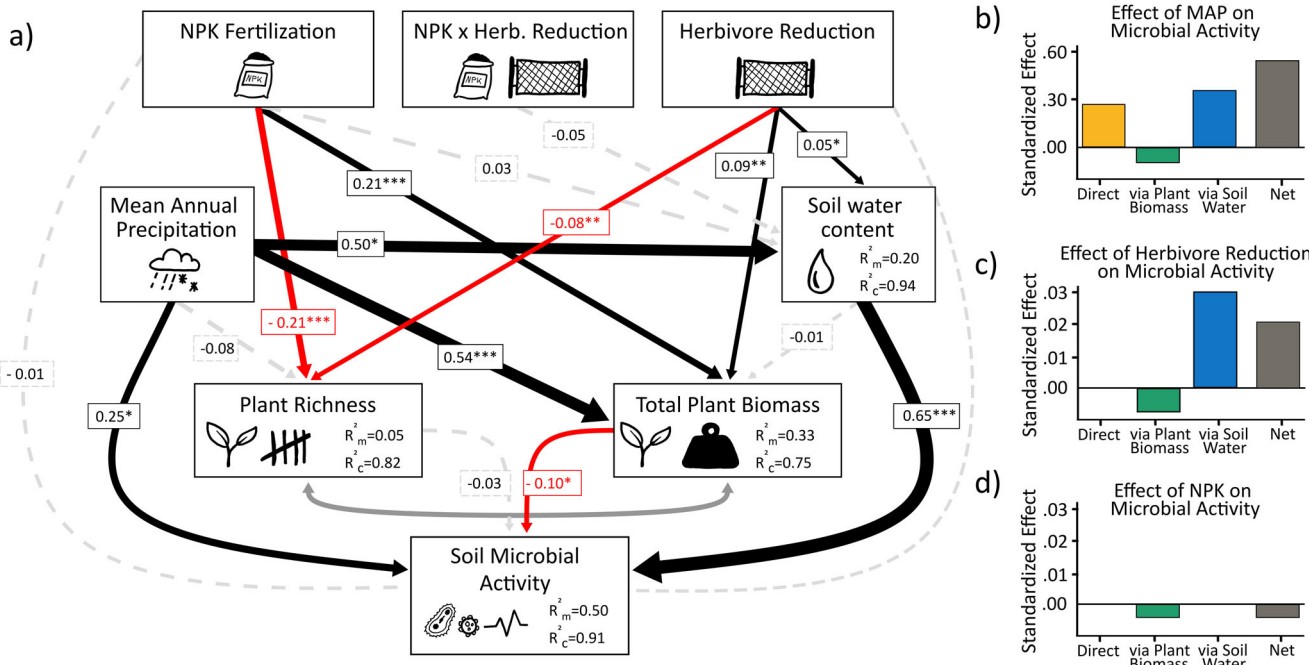

**Fig. 3 Structural equation model of soil microbial activity. a** *Soil microbial activity as a best-fit Structural Equation Model showing the effects of NPK fertilization, herbivore reduction (A/C = 77.9, Fisher's C = 1.932; P = 0.381; d.f. = 2; 26 sites). Black arrows indicate significant positive and red arrows indicate significant negative effects in the model (P < 0.05). Dashed gray arrows indicate non-significant effects (P > 0.05) that remain in the model based on AIC. Dark gray double-headed arrows indicate paths that were treated as correlated errors in the model. Arrow widths are proportional to their effect sizes. Numbers along the arrows are standardized path coefficients. Marginal R²ₘ: model variation explained by fixed effects; conditional R²c: model variation explained by both fixed and random effects. Significance levels: *p < 0.05; **p < 0.01; ***p < 0.001.* **b** Direct, indirect, and net effect of MAP on soil microbial activity, and **c** direct, indirect, and net effect herbivore reduction on soil microbial activity, and **d** direct, indirect, and net effect of NPK fertilization on soil microbial activity (scale of b) differs from **c** and **d**.

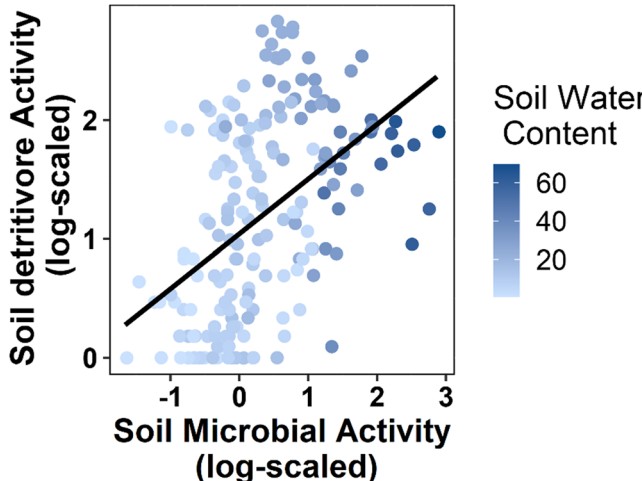

**Fig. 4 Correlation between soil microbial and detritivore activity.**
Correlation of soil microbial activity and detritivore activity (both log-scaled, data from 18 sites included; $F = 9.15$, $p = 0.003$). Color of data points (blue) indicates soil moisture level of the sample.

their ability to move to deeper soil layers[65]. This adaptability allows them to sustain their activity even in drier conditions. In support of this, Sagi et al.[66] discovered that the primary litter decomposition in the Negev desert during summer was driven by a woodlice species, in contrast to microbes which lacked the necessary water for growth.

Other possible mechanisms for direct negative impacts of (especially larger) herbivores on soil detritivores and thus the positive effect when they were excluded, entail physical disturbances like trampling and soil compaction[53,67,68]. These result in higher bulk density and reduced connectivity of soil pores[69] that normally ensure water infiltration and air permeability[70,71]. Such a reduction in soil pore space has been shown to reduce the abundance and diversity of soil arthropods and annelids[69,72,73]. For example, Collembola and enchytraeids strongly depend on macropores in their living environment, have hardly any ability to move through compacted soil, and may thus experience reduced access to food resources, consequently inhibiting their feeding activity[69]. However, even soil animals with considerable burrowing abilities, like earthworms, have been shown to be negatively affected by soil compaction[74]. Indeed, we found some evidence for a significant positive relationship between soil microbial and detritivore activity and soil porosity across a subset of sites (Fig. S8). However, given that only a subset of the sites could be considered for this analysis, this topic needs to be addressed in future research.

At the same time, we observed increases in plant biomass that were associated with herbivore reduction, nutrient addition, and higher levels of precipitation[75]. It is well-established that vegetation cover helps to maintain high levels of soil biological activity, as evidenced by previous studies[76–80], which is consistent with our own findings. However, higher plant biomass also led to declines in soil microbial activity which is, in contrast to other studies reporting positive effects of higher plant biomass on soil biological activity *via* bottom up effects of increased rhizodeposition[19,81–83]. There are several possible explanations. On one hand, enhanced plant growth could potentially result in higher transpiration rates[84,85], ultimately leading to a reduction in soil water content over time. On the other hand, herbivore reduction also led to a less diverse plant community, which could also decrease microbial respiration. It is also possible that soil microbial communities have to compete with plants for nutrients, possibly leading to reduced respiration rates. Follow-up studies are needed to relate environmental change-induced alterations in soil microbial communities to ecosystem functions, using standardized, replicated methods to increase the generality and robustness of such experiments[86,87].

In contrast to our hypotheses (2 & 3), we did not observe an either significant effect of nutrient addition soil biological activity or a significant interaction between nutrient addition and herbivory. Although we applied NPK at high levels, we did not detect any direct fertilization effect, suggesting that availability of mineral nutrients is not the main determinant driving soil biological activity in grasslands. However, soil community responses and functions can be diverse and context-dependent[88]. Previous studies have shown that nutrient addition can alter the soil community by changing pH, porosity, organic fractions, and increasing decomposition, but responses of soil microbial respiration and biomass to NPK addition are highly variable[25,40,41,89–91]. However, our findings show a decline in soil microbial activity due to nutrient addition in global grasslands, alongside an increase in total plant biomass. Nutrient addition altered plant communities through by increasing total plant biomass and reducing plant species richness. Similar effects have been reported by multiple studies[46,92–97]. As nutrient addition has been shown to decrease soil organic matter stabilization, we

determine whether this pattern is causal or due to covariation with other variables such as geological and historical factors, it will be necessary to conduct future experimental manipulations. Both soil biological activity measures are tightly linked to decomposition processes that determine carbon sequestration and release[11,51], and therefore play a key role in grassland carbon cycles[52]. In addition, herbivore presence was associated with lower soil moisture which may amplify effects of a drier climate. Moreover, MAP, NPK addition, and herbivore reduction had indirect negative effects on soil microbial activity by increasing plant biomass. This indicates that plant community-mediated changes in soil microbial communities and functions depend on abiotic and biotic conditions. This study improves our mechanistic understanding of factors determining soil biological activity globally, which is crucial to predict belowground ecosystem functioning in a changing world and to adopt measures to preserve grassland systems[2,46,53,54].

In line with our hypothesis (1), we found a consistent overall positive effect of herbivore reduction on soil detritivore activity across all sites. In addition, through our SEM analysis, we observed that herbivore reduction enhanced soil detritivore activity directly and also indirectly via an increase in soil water content. However, while herbivore reduction had no direct effect on soil microbial activity, it also indirectly increased microbial activity via increases in soil water content. It has long been recognized that soil organisms strongly depend on soil moisture (e.g., residing on water films within the soil pore system)[11,55–58]. Confirming this, our results highlight the role of water availability for both measures of soil biological activity, with higher activity levels at sites with higher MAP and soil water content. Previous studies have found herbivores to reduce soil water content[59] and to have negative effects on soil organisms, especially in unproductive ecosystems[17,19,53,60,61]. Our results suggest that a reduction in soil activity with herbivory is the dominant pattern in grasslands. This finding is consistent with other studies reporting decreased soil respiration in response to lower soil water content[62–64]. In our study, soil water content had a significantly larger effect on soil microbial activity compared to detritivore activity. This might be attributed to the fact that the soil water content data were better aligned with the measure of soil microbial activity. However, it is also possible that detritivores are less influenced by water content than microorganisms due to

speculate that our results may be connected to shifts in plant or soil microbial communities that influence the recalcitrance of organic matter and its microbial processing[98].

Assessing soil microbial activity for a global scale study entailed some constraints that could have also influenced our findings. Measuring soil microbial activity involved homogenizing, sieving, and shipping soil samples to a central laboratory, which includes disruption of soil aggregates, along with potential changes in microbial activity due to shipping conditions. Additionally, assessing microbial activity under controlled temperature conditions, differing from the natural environment, might have influenced the correlations observed with mean annual temperature (MAT). Similarly, the bait lamina test, offering an on-site approach to assess detritivore activity, has faced recent critique regarding its use of standardized substrates as accurate indicators of local plant litter-driven decomposition[99] even though this concern was not directly tested for bait lamina strips.

Overall, our results highlight that reductions in mean annual precipitation or prolonged drought periods may reduce soil biological activity that is key for the provisioning of essential ecosystem functions like nutrient cycling and decomposition[52,100]. These expected changes in climate could be further amplified by alterations in the abundance and identity of herbivores as well as nutrient inputs, with complex feedback mechanisms, including shifts in local plant community composition and productivity, as well as abiotic factors like soil compaction. Nutrient addition did not directly affect soil biological activity across global grasslands, emphasizing the importance of an indirect effect via plant biomass that should be considered in future studies. These novel insights into the global drivers of soil biological activity stress the complex interplay between different components of anthropogenic change that may alter above-belowground interactions and thus the functioning of global grasslands.

## Methods

**Experimental design and included sites**. Fieldwork was conducted in 2015 within the Nutrient Network Global Experiment (www.nutnet.org)[46] at 26 sites (see Fig. 1a and Table S1). The sites are located in North and South America, Europe, Asia, and Australia and are all dominated by herbaceous or low-statured vegetation (hereafter referred to as grasslands). Moreover, sites cover wide environmental gradients with elevations ranging from 6 m to 4.241 m a.s.l., mean annual temperatures from −3.3 °C to 22.4 °C, and mean annual precipitation from 324 mm to 1678 mm (Table S1). The experiments were set up at different times in 2009–2014 (for details see Table S1).

For our study, we sampled plots at each site, which were randomly assigned to one of four: Control, nutrient addition, fenced, nutrient addition and fenced. Each treatment was replicated three times at each site, leading to a total of 12 plots. Fences excluded aboveground herbivores weighing more than 50 g. The plots were 5 × 5 m in size and NPK plots received a fertilization treatment of nitrogen (N), phosphorus (P), and potassium (K). Nutrient addition rates and sources are: 10 g N m$^{-2}$ * year$^{-1}$ as timed-release urea [(NH$_2$)$_2$CO], 10 g P m$^{-2}$ * year$^{-1}$ as triple-super phosphate [Ca(H$_2$PO$_4$)$^2$], 10 g K m$^{-2}$ * year$^{-1}$ as potassium sulphate [K$_2$SO$_4$]. Additionally, 100 g m$^2$ of a micronutrient mix of Fe (15%), S (14%), Mg (1.5%), Mn (2.5%), Cu (1%), Zn (1%), B (0.2%), and Mo (0.05%) were applied once at the start of the experiment. In contrast, control plots did not receive additional nutrients and represent ambient soil conditions. The fences for herbivore reduction were of 2.3 m height, with few sites having physical constraints that required fence modification. They were set up with a 1 cm woven wire mesh extending 0–90 cm aboveground and a 30 cm outward-facing flange stapled to the ground to exclude smaller digging animals. To reduce possible impacts of neighboring plots, all plots are separated by walkways of at least 1 m width. All sampling occurred in a single, randomly selected, 2.5 × 2.5 subplot of each plot. Further details on the experimental set up, standardized sampling protocols, and nutrient sources are described in Borer et al.[46].

**Plant data**. Following the standardized Nutrient Network protocol, total aboveground plant biomass was clipped at peak biomass within two 0.1 × 1 m strips per plot, whose locations are changed each year (see Borer et al., for details). Sorted plant material was dried at 60 °C to a constant mass and weighed to the nearest 0.01 g. For our analyses, we used data on total plant biomass (i.e., the sum of dead and live plant biomass) from 2015 (i.e., the year of the study) as a proxy for plant-derived inputs to the soil, such as rhizodeposits and plant litter. Plant species richness was assessed on-site in a permanent 1 × 1 m quadrat located in the focal subplot in each plot.

**Climate variables**. Data on mean annual precipitation (MAP in mm) and mean annual temperature (MAT in °C) were derived from the WorldClim database (version 1.4; Hijmans et al.). Values were interpolated at high resolution from meteorological stations with 10 to 30 years of data[101].

**Soil sampling**. Soil invertebrate feeding activity was assessed at 16 sites using the bait lamina test (Terra Protecta GmbH, Berlin, Germany), which is commonly used as a rapid ecosystem function assessment method[11,47]. The bait strips are made of PVC (1 mm × 6 mm × 120 mm) and have 16 holes (1.5 mm in diameter). Holes were filled with an artificial organic bait substrate, which was prepared according to the recommendations of Terra Protecta, consisting of 70% cellulose powder, 27% wheat bran, and 3% activated carbon. The bait substrate is primarily consumed by soil collembolas, enchytraeids, and earthworms[76,102]; microbial activity plays a minor role in bait loss[103–105]. The bait strip assessment was done by the principal investigator of each site. The bait strips were inserted vertically into the soil with the uppermost hole just beneath the soil surface. A steel knife was used to create a slot in the soil, before the strips were inserted. Five strips were spaced 15 cm apart within each plot to account for within-plot spatial heterogeneity. After three to six weeks of exposure, the bait strips were removed from the soil and directly evaluated in the field. Each hole was rated as 0 (no invertebrate feeding activity), 0.5 (bait material partly consumed), or 1 (bait material completely consumed), based on visual inspection. Thus, soil invertebrate feeding activity could range from 0 (no feeding activity) to 16 (maximum feeding activity) per strip. Mean bait consumption of the five strips was calculated per plot prior to statistical analyses and expressed as a percentage. Timing variations resulted from the substantial environmental differences, as in some cases, short exposure intervals did not yield discernible changes (for detailed exposure time please see Table S1).

Soil for microbial data was collected from 26 sites six weeks before peak plant biomass production (local site coordinators chose specific dates, as seasonality varied across different latitudes) by taking three subsamples per plot (using a soil corer with 5 cm diameter and 12 cm depth), which were then homogenized and sieved using a 2 mm mesh. All soil samples to Anita Risch in Switzerland following a standardized protocol. A subset of these samples was then shipped to a centralized lab at the German Centre for Integrative Biodiversity Research in Leipzig, Germany. We ensured sample quality during transit by using postal services with temperature control and fast shipping methods. Here, we took approximately 6 g of fresh soil to measure

basal respiration (without the addition of substrate) at hourly intervals for 24 h at 20 °C using an $O_2$-microcompensation system[48]. We used four different $O_2$-microcompensation devices to measure all samples simultaneously. Basal respiration, as a measure of soil microbial activity, was then calculated as the mean $O_2$ consumption rate 14–24 h after the start of the measurements (µl $O_2$ h$^{-1}$ per g soil dry weight), as the machine needs some time to measure stable values over an extended period[77]. In addition, soil water content [%] was calculated as the difference between the weight of the fresh soil sample and the weight of the soil sample per plot after they were dried for at least 48 h at 70 °C. Soil water content was significantly positively correlated with soil water holding capacity ($R^2 = 0.61$, $p < 0.001$). Soil porosity was determined as described in Risch et al.[25], but was only available for a subset of 15 sites.

**Statistics and reproducibility**. To assess the effects of fertilization, reduced density of vertebrate herbivores, and their interaction on soil detritivore feeding and microbial respiration across all sites without accounting for abiotic factors and plant data, we employed linear mixed-effects models using the lmer function from the R-package "*lme4*[106]". The model's random intercepts were organized based on two factors: (1) block nested within site, and (2) the type of $O_2$-microcompensation device (for soil microbial data) or field exposure duration (for detritivore data), as some sites had a longer exposure time due to logistical constraints. We also tested a model with treatment duration years as a fixed effect, but found that treatment duration had no significant impact on soil microbial and detritivore activity, and consequently excluded "treatment duration" from our explanatory parameters. To account for the non-normality of our response variables, we log-transformed data prior to our analysis. Figure 1b, c are based on mixed-effects model fits extracted using the package "*ggeffects*[107]".

We used structural equation modeling (SEM) to disentangle direct and indirect pathway effects by which fertilization and herbivore reduction affected the activity of soil organisms. In determining the environmental variables for our SEM approach, we considered factors that could offer meaningful insights into the dynamics of soil microbial and detritivore activity. These variables were selected a priori based on their established influence on soil microbial and detritivore activity, as well as their potential to mediate treatment effects. Our choices were guided by existing literature in the field. Given that plant community composition and biomass are strong predictors of soil microbial and detritivore activity and are thus likely to mediate the treatment effects, we included plant species richness and total plant biomass in the SEM (Figs. 2 and 3). As soil organisms are highly dependent on soil moisture[108], we included soil water content as a key abiotic driver. We also chose to include mean annual precipitation (MAP) as another exogenous variable, as it was correlated with soil moisture (Fig. S7) with the two soil activity variables (Figs. 2 and S4a, b) and should have long-lasting effects on soil conditions that are also relevant for our snapshot assessments. We further selected mean annual temperature as another exogenous variable. However, the relationship between mean annual temperature (MAT) and microbial activity was not statistically significant ($p = 0.14$), and a similar non-significant trend was observed with detritivore activity ($p = 0.93$) (see also Fig. S4c, d). Although MAT displayed a positive correlation with plant species richness, this association did not extend to the other variables in our SEM model (as illustrated in Figs. 2 and 3). Consequently, we decided to exclude MAT from the final model. The framework of the "*piecewiseSEM*" R package[109] allowed us to test for interactive treatment effects and to account for the

hierarchical study design by including random effects in the models. We also investigated the effects of soil pH and individual effects of living and dead plant biomass on soil microbial and detritivore activity within the model. However, these data were only available for a small subset of sites and as we found no significant direct or indirect effects on biological activity, we excluded them in the final model.

The single models that were incorporated in the SEM were built using LMMs (Table S2). The assumptions of the LMMs were checked by plotting frequency distributions of each variable and the variance structure of all models using residual plots for homogeneity and quantile-quantile plots for normality (i.e., no correlation between the residuals and the fitted parameters of the model). To meet model assumptions, plant biomass, plant species richness, soil water content, and detritivore activity were log-transformed. The relationship between plant richness and total plant biomass was included as a correlated error term due to reciprocal effects[110].

The number of variables was reduced from the conceptual model using the Akaike Information Criterion (AIC) that is implemented in the "*piecewiseSEM*" package. Standardized coefficients are reported for each path of the final model (Tables S3 and S4, Figs. 2a and 3a). The overall fit of the models was evaluated by using Shipley's test of d-separation obtained through Fisher's C statistic. Correlations were performed between soil microbial activity and soil detritivore activity. To examine the impact of herbivore-induced changes in soil structure on biological activity, we analyzed the correlation between soil porosity (a measure of soil compaction influenced by herbivores) and the two soil activity measures. However, due to insufficient sample size, we could not include soil porosity in the SEM. The statistical analyses were performed using the R statistical software (version 4.2.2.; R Core Team 2022). Data used for creating the figures can be found in the supplementary data (Figs. 1a, 2 and 4 were created with Data 1, Figs. 1b and 3 with Data 2).

**Reporting summary**. Further information on research design is available in the Nature Portfolio Reporting Summary linked to this article.

## Data availability
The source data that support the findings of this study can be found in the supplementary data (Figs. 1a, b, 2 and 4 were created with Data 1, Fig. 1c and Fig. 3 with Data 2). All other data are available from the corresponding author on reasonable request.

## Code availability
The code is available from the corresponding author upon request.

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

## Acknowledgements

This work was generated using data from the Nutrient Network (http://www.nutnet.org) experiment, funded at the site-scale by individual researchers. Coordination of soil sampling was funded by a competitive WSL internal grant to A.C. Risch and S. Zimmermann. Coordination and data management for NutNet have been supported by funding to E. Borer and E. Seabloom from the National Science Foundation Research Coordination Network (NSF-DEB-1042132) and Long-Term Ecological Research (NSF-DEB-1234162 to Cedar Creek LTER) programs, and the Institute on the Environment (DG-0001-13). We also thank the Minnesota Supercomputer Institute for hosting project data and the Institute on the Environment for hosting Network meetings. J. Siebert, M. Sünnemann, and N. Eisenhauer acknowledge funding from the German Centre for Integrative Biodiversity Research (iDiv) Halle-Jena-Leipzig, funded by the DFG (FZT 118). M. N. Bugalho thanks the Portuguese Foundation for Science and Technology (FCT) for funding through contract DL 57/2016/CP1382/CT0030 and projects UID/BIA/50027/2013 and POCI-01-0145-FEDER-006821. M. N. Bugalho also thank Rui Alves for granting access to the study site (comp.pt) We acknowledge the Portuguese Science Foundation (FCT) for funding the research unit CEF (UIDB/00239/2020). We thank Felix Gottschall for support with Figs. 2 and 3 and especially the design of the icons.

## Author contributions

N.E. conceived the study; A.C.R and N.E. developed the idea of a joint add-on project within the Nutrient Network. E.T.B. and E.W.S. coordinate the Nutrient Network. A.C.R., S.Z., and J.S. coordinated the global sampling campaign; J.D.B., L.B., D.M.B., E.T.B., M.N.B.; A.A.B.B., M.C.C., E.C.; K.F.D.; A.E.; N.H., J.M.H.K., A.S.M., R.L.M., J.L.M., S.A.P., J.N.P., E.W.S., R.S. and C.J.S. contributed data; J.S. and Y.H. analyzed the data; J.S., M.S., and N.E. wrote the manuscript with input from all authors. M.S. revised the manuscript with input from all authors (Table S5).

## Funding

## Competing interests

The authors declare no competing interests.

## Ethical approval

We support inclusive, diverse, and equitable conduct of research.

## Additional information

[1]German Centre for Integrative Biodiversity Research (iDiv), Halle-Jena-Leipzig, Puschstrasse 4, 04103 Leipzig, Germany. [2]Institute of Biology, Leipzig University, Puschstrasse 4, 04103 Leipzig, Germany. [3]Ecology and Biodiversity Group, Department of Biology, Utrecht University, Padualaan 8, 3584 CH Utrecht, The Netherlands. [4]Swiss Federal Institute for Forest, Snow and Landscape Research WSL, Community Ecology, Zuercherstrasse 111, 8903 Birmensdorf, Switzerland. [5]School of Environmental and Forest Sciences, University of Washington, Seattle, WA 98195, USA. [6]Department of Ecology, Evolution, and Organismal Biology, Iowa State University, Ames, IA 50010, USA. [7]USDA-ARS Rangeland Resources & Systems Research Unit, Fort Collins, CO 80526, USA. [8]Department of Ecology, Evolution, and Behavior; University of Minnesota, St. Paul, MN 55108, USA. [9]Centre for Applied Ecology "Prof. Baeta Neves", School of Agriculture, University of Lisbon, Tapada da Ajuda, 1349-017 Lisbon, Portugal. [10]Department of Earth and Environmental Sciences, The University of Manchester, Oxford Road, Manchester M13 9PT, UK. [11]Forest Research Centre, School of Agriculture, University of Lisbon, Lisbon, Portugal. [12]Ecology, Behavior and Evolution Section, University of California San Diego, 9500 Gilman Dr. #0116, La Jolla, California 92093-0116, USA. [13]Department of Ecology and Evolutionary Biology, University of Colorado, Boulder, CO 80309, USA. [14]Ecology and Genetics Unit, University of Oulu, P.O. Box 8000, FI-90014 University of Oulu, Oulu, Finland. [15]Helmholtz Center for Environmental Research – UFZ, Department of Physiological Diversity, Permoserstrasse 15, 04318 Leipzig, Germany. [16]Mammal Research Institute, Department of Zoology & Entomology, University of Pretoria, Pretoria, South Africa. [17]Health & Environmental Sciences Department, Xi'an Jiatong-Liverpool University, Suzhou, China. [18]Department of Integrative Biology, University of Guelph, Guelph, Ontario N1G 2W1, Canada. [19]Department of Plant & Soil Sciences, University of Kentucky, Lexington, KY 40546, USA. [20]Arthur Rylah Institute for Environmental Research, 123 Brown Street, Heidelberg, VIC 3084, Australia. [21]School of Biological Sciences, Monash University, 25 Rainforest Walk, Clayton, VIC 3800, Australia. [22]Hawkesbury Institute for the Environment, Western Sydney University, Locked Bag 1797, Penrith, NSW 2751, Australia. [23]School of Agricultural, Environmental and Veterinary Sciences, Charles Sturt University, Albury, NSW 2640, Australia. [24]Harry Butler Institute, Murdoch University, 90 South Street, Murdoch, WA 6150, Australia. [25]Institute of Agriculture, The University of Western Australia, 35 Stirling Hwy, Crawley, WA 6009, Australia. [26]Lancaster Environment Centre, Lancaster University, Lancaster LA1 4YQ, UK. [27]Swiss Federal Institute for Forest, Snow and Landscape Research WSL, Forest Soils and Biogeochemistry, Zuercherstrasse 111, 8903 Birmensdorf, Switzerland. [28]These authors contributed equally: Julia Siebert, Marie Sünnemann. ✉email: marie.suennemann@idiv.de

