## [Peer Review File · Communications Biology]

Drivers of soil microbial and detritivore activity across global grasslandsReviewers' comments:

Reviewer #1 (Remarks to the Author):

Review of the manuscript: "Drivers of soil biological activity across global grasslands"

In this manuscript, Siebert & Sünneberg et al. present the results of an impressive coordinated study on the response of soil microbial and detritivore activity to nutrient addition and herbivore reduction in 26 grasslands across 5 continents. They report that the response of the activities of both groups to nutrient addition and herbivore reduction was mediated by the change in soil water content, indicating that soil water content is the main driver of soil biological activity.

Overall, this is a well-executed study and a rather well-written manuscript. This research is timely, particularly on the global drivers of detritivore activity which is largely understudied, relative to that of soil microbial activity. I made a number of minor comments while reading the manuscript (see at the bottom), but most importantly, I have a few general comments which I think could improve the manuscript if integrated during the revisions. These include comments on the introduction of the study, on the reference throughout the manuscript to literature on the biology of the organisms of interest, on the use of climatic factors, on the lack of plant traits, and on acknowledging the limitations of the present study.

Main comments:

1. The introduction: I am struggling a bit with the introduction, which I find lacking structure and somewhat confusing. The second paragraph starts talking about N and P depositions in the first line and then switches to herbivory for the rest of the paragraph with no further mention of nutrient inputs, with no knowledge gap identified, and without prior information on the importance of soil organisms. The third paragraph introduces the importance of soil organisms at last, and the last paragraph introduces the response of soil/soil organisms to nutrients additions. I would encourage the authors to rethink the structure of the introduction to improve its logic. After briefly thinking about it, I would suggest:

- Paragraph 1: largely unchanged
- Paragraph 2: introduce the importance of soil microorganisms and detritivores to soil processes, and potential mediators of global changes. Mention the known drivers + identify the lack of knowledge on unknown responses to herbivore pressure and nutrient addition.
- Paragraph 3: introduce the importance of herbivory and the knowns and unknowns on the response of soil biological activity
- Paragraph 4: introduce the importance of nutrient additions (direct effect and indirect through plant responses).
- Paragraph 5: introduce the study

2. The biological context: Overall, throughout the manuscript (and in the introduction in particular), much of the focus is on the broad factors of interest (nutrient input, herbivory), with little information on the response variables: the activity of the organisms of interest. There is in fact substantial information in the literature on the response of soil biological activity (albeit not at the global scale of the present study). It would be good to see this information in this manuscript: Who are these organisms? What do they do? What are detritivores? Why focus on both microbial and detritivore activity? What do they both do respectively for soil processes? What do we know about what drives their activity, from lab and field studies? Why might they respond differently to the drivers mentioned, and the factors of interest? Overall, the study and its interpretation would benefit from a more thorough presentation of the literature on the biology/ecology of soil organisms.

3. The effect of MAP / Soil water content: I am a bit confused about the climatic factors. Was the soil water content measured at the time of soil collection for soil microbial activity, and if so, how representative is it of the soil water content during the period of bait lamina field incubation? Also, I am not quite sure about the process for building the SEM, with the factors included or not after preliminary data analyses rather than after building an a priori model based on the current knowledge.

Is the fact that MAT doesn't correlate with activity a good reason to remove it from the SEM? Perhaps it has a negative effect on something that has a positive effect, leading to no visible effect. In fact, I would encourage the authors to consider the aridity index (ratio of MAP to MAT) as another climatic variable which integrate both MAP and MAT. MAP on its own is not necessarily a good proxy of available water to soil organisms, as this depends on the level of evapotranspiration. MAP in London (580 mm) is less than that of Paris (720 mm) which is less than that of Rome (870 mm), yet I wouldn't bet that Rome has a higher soil water content than London ;-)

Regardless of the climatic factors used, I was missing a discussion on why the activity of detritivores and microbes is controlled more a less tightly by climatic variables. It may be that the soil water content data was more aligned with the soil microbial activity measure, but it may be that detritivores are less controlled by water content than microorganisms. Detritivores, by being able to move, can find water in deeper soil or moist pockets in the soil, allowing them to maintain an activity even in drier conditions. In line with that, Sagi et al. 2019 (Proc. Roy. Soc. B) found that most of the litter decomposition in the Negev desert in the summer was carried out by a woodlice species, not by microorganisms which didn't have enough water to grow. Interestingly, in Fig 4 it appears that soil detritivore activity is less variable (can this be inferred from log-log relations?) than soil microbial activity, and peaks at moderate levels of microbial activity and soil water content. This figure also suggest that the relation between detritivore activity and soil water content is not linear.

4. Plant trait data: One key parameter that would really help understand the response of soil biological activity to these factors is plant trait data. Indeed, this resource quality is likely affected by changes in NPK / herbivory, with important consequences for soil biological activity. This is mentioned in the introduction but not considered later on. I am not suggesting that the authors change their SEM to include it at this stage, but it would be good to read their thoughts on it. This may help explain the negative relationship between plant biomass and microbial activity.

5. Limit acknowledgments: The methods used to assess microbial and detritivore activities are somewhat coarse methods. Their limitations should be acknowledged and the conclusions should be nuanced based on these limitations. Soil microbial activity was measured on soil samples that were sampled, homogenised, sieved, shipped (how long did it take and under which conditions?), and O₂ consumption measured in a centralised lab. This post-processing (disruption of aggregates), shipping (continued microbial activity under altered conditions) and ex-situ measure of soil microbial activity under constant temperature likely led to significant biases. This may also explain part of the lack of correlation between microbial activity and MAT. In turn, the bait lamina, although allowing to measure detritivore activity in situ, also comes with limitations. It was recently shown that the disappearance of standardised substrates across multiple sites (as is the case here), as indicator of soil biota-driven decomposition, is a poor indicator of the soil-biota driven decomposition of the local plant litter (under realistic conditions; Joly et al. 2023, Nature Ecology and Evolution). This questions the relevance of data obtained from standardised substrates, and the validity of the derived conclusions. This was not shown specifically for bait lamina, but it raises concerns about this method. This is not to discredit the present study, but it would be good to see a section in the discussion bringing up possible limitations of this study. Note that the descriptions of these methods lack details (e.g., transport of soil for microbial activity; period during which the bait lamina were inserted; etc.).

Line 119: 'studied' rather than 'studies'?

Line 119: what do you mean by "macro-biota"? The associated citations rather refer to microorganisms, a bit to soil fauna, or to arthropods in general but not specifically soil arthropods (Lind et al. 2017; not necessarily a relevant citation here?).

Line 121: It would be good to see more specific citations to the wealth of work on the function carried out by soil animals rather than broad chapters/reviews on soil biodiversity that are not necessarily function-focussed.

Line 128: This is the first time that you mention invertebrate detritivores. Can you describe what they are/do? Some of the taxa that you refer to later in the paper are in fact rather microbial grazers (collembola, mites) or predators (mites), rather than detritivores.

Line 129: Why mention micronutrients when they are not discussed afterwards, with only mentions of NPK?

Line 190: '(>50g)' What does this refer to? I don't know what to make of it...

Line 227: woodlice, no? I am not sure mites or collembola would directly feed on the substrate – they are rather microbial grazers...

Lines starting 220: when was this done, and why the variable time of exposure (3 to 6 weeks)?

Lines starting 239: when was this done relative to the bait lamina tests? How were the soil samples transported to the centralised lab?

Line 373: you mean herbivore *presence* was associated with lower soil moisture?

Line 413: "reduce"

Discussion: It is somewhat confusing, as some parts discuss the response of detritivore activity and then generalise to "biological activity", without a parallel discussion of the response of microbial activity. Similar to the introduction, I am a bit lost in the logic of this section.

Line 452: Things are immobile or they are not – it is not a gradual state. Less mobile? In any case, detritivore do not have "host plants", so these references to plant parasites/herbivores is not relevant.

Line 457: why refer to AMF (microorganisms) when this paragraph discusses the response of detritivores?

Line 461: does this refer to detritivore activity, or microbial activity, or both? This should be a lot clearer, here and throughout the discussion.

Fig. 2: Ouch! The icon in the box 'soil detritivore activity' is a centipede (one pair of leg per segment, and modified legs at the back). Centipede are not detritivores, but predators. Please change to a millipede for instance (two pairs of leg per segment and no modified legs at the back) – many icons are available on line for inspiration.

Reviewer #2 (Remarks to the Author):

COMMSBIO-23-1077

Siebert et al. conducted an interesting global experiment to evaluate which drivers structure detritivore and microbial activity in grasslands. The experiment included two treatments: (1) addition of nutrients and (2) reduced grazing. In addition to these treatments, they also included other predictors such as plant biomass and species richness and soil water content, which were in turn analyzed as responses of other predictors with Structural Equation Models (SEMs), aiming to disentangle direct and indirect effects on detritivore and microbial activity. They found that detritivore activity, but not microbial activity was affected by reduced grazing. The addition of nutrients had in both cases no direct effect, but was found to indirectly affect both detritivore activity through regulating plant biomass and plant species richness.

Overall, I think the manuscript is well written, especially the discussion, the study is interesting and relevant. Some aspects of the sampling and analysis were not clear to me and I suggest that the authors improve this. I explain this in more detail below. But apart from these 'minor revisions', I think the paper is well developed and can be published.

General comment:

Regarding the sampling design, there is unclarity about the treatment duration, and sampling times. It can be understood from lines 232-233 that the invertebrate feeding activity was measured within 3 to 6 weeks, from lines 240-243 that the soil was collected 6 weeks before peak plant biomass production and from lines 261-266 that treatments were in place for different lengths of time. However, it is not clear from the text how long these treatments were in place and how the sampling related to season (although collecting 6 weeks before plant biomass peak suggests seasonal adjustment of sampling). Apparently, the treatments were sometimes in place for multiple years. Please clarify sampling times, treatment lengths and the relation with season.

Specific comments:

line 119 "While they have been little ..." -> "While there have been few ..."

Line 196-197 "... plots do not ..." -> "... plots did not ..."

Line 211-213 Perhaps state instead of "visually estimated" that plant species richness was counted on

site. I could not find further details on this in Borer et al (2014b). Generally, the authors refer quite often to this study for more details, but it is a very compressed Nature article. I suggest the authors double check if these details are indeed easily found in Borer et al. (2014b).

Line 232-233 Table S1 has a column with the year that the experiment started. I suggest to also add the exact date to Table S1

Line 240-243 While this suggests sampling was adjusted to season, it would also for microbial activity be good to add the sampling dates to tables S1.

Line 257-260 Are these separate models that only include NPK x Reduction or are these the models presented as part of the SEM in Tables S3 and S4? Please clarify.

Line 261-268 This section is somewhat confusing. The second sentence seems to repeat what was stated in the first sentence ("... (2) the number of years the treatments had been in place at a specific site to account for differences among sites ..."). But the way it is formulated sounds as if it concerns another random effect. Also, the statement in the first sentence is confusing, as the differences among sites was already corrected for by including site itself as a random effect. I think the authors mean they included this to adjust for duration time of the experiment? Finally, (lines 266-268) suggest that duration time was first included as a fixed effect and then excluded as it was not informative. I think this is more appropriate as this would allow to model it as a continuous variable, which would preserve the putative relationship with time, instead of turning each year into an independent level as is done in the case of the random intercepts. If it is then deemed not informative based on the AIC criterion or p-value, the variable can be ignored and does not have to be included as random effects. I also wonder if rounding time length to year is optimal and if it would not be better rounded to months or days. Please clarify this rationale behind the decisions taken for the analysis.

line 315-316 It could be interesting to also calculate this correlation in the based on the residuals of the detritivore activity and microbial activity models, because as the color scale in Figure 4 also suggest this correlation is probably mainly driven by water content and it might be interesting to see how the responses are correlated after controlling for the effects of the studied variables.

Figures Generally, the figures are very pixelated and in some cases difficult to read. It is recommended to present vector figures.

Fig. 1 Are the presented data points raw observations or partial residuals?

Reviewer #3 (Remarks to the Author):

The manuscript shows the results of the global, well-designed experimental set up NutNet where effects on nutrient addition and herbivore reductions have been monitored. The submitted study disentangles the effects and mechanisms of both treatments on soil biological activity finding that despite no direct treatment effects, treatment effects were mediated via their influence on soil water content and plant biomass. I find this study well-written and analysed showing the processes and context-dependency behind global change drivers (eutrophication, grazing) on soil microbial activity i.e., the base for ecosystem functioning.

I have only minor comments on the presentation of the introduction, statistical methods, the results, and parts of the discussion.

Introduction:

L106: Here you mention changes in herbivore densities. You look at a reduction treatment, how does this fit together with replacing wild by domestic herbivores. Reading this one could think that herbivore pressure globally is not reduced then and one should look rather on effects of increased herbivore pressure. How could you put this statement of shifts in herbivores (wild vs domestic) into the context of your treatment?

L129: I was struggling in this section on nutrient stoichiometry to see it's "added value". All the information is relevant but partly I felt it repeats things said in the sections before. Was it to mention

context-dependent effects of grazing, grazing mechanisms? Maybe just a better topic sentence is missing?

Statistical methods:

When introducing the SEM approach, the selection of environmental variables was not very clear. I was missing some more overview about selection criteria for environmental variables, why e.g., soil organic carbon was not included which is known in other studies as an important driver. You mention this in your discussion, too (L409). Probably this can be simply solved by restructuring this section a bit, mentioning the selection of environmental covariates and reason for their selection in the beginning of the section. Did you base the selection on previous findings or on correlation of those parameters with the response variables? Information like this I was missing. This would give the reader simply a better overview of the logic of you SEM.

Finally, in the abstract it was mentioned that soil biological activity was measured on 28 and detritivore activity only on 16 grasslands. Please, mention the different sample sizes in the methods section again.

Results:

The figure captions of the SEM do not introduce the content in the three different panels. This could be solved with one sentence telling what is shown in panel a, b and c.

L325 How can it be that the LMM shows no effect of herbivory, but the SEM does show a direct effect?

Discussion:

You start the discussion with the topic of soil moisture (L381-394), which is the most prominent in your results, I agree. Yet, this topic occurs within the other sections again, as it's the cause for indirect effects of the main factors - fertilization and herbivory. To avoid this redundancy, I suggest to - after the opening paragraph - start straight away with the sections about hypothesis 1 and 2 and include the explanation on soil moisture within them.

In relation to hypothesis 2 (L418...) you mention that it was partially confirmed through the direct herbivory effects on detritivore activity. As the hypothesis includes also soil biological activity, I was missing at least one sentence mentioning that this effect was not found - or only in a direct form. Otherwise, I was stuck wondering about why the hypothesis was only partially confirmed.

Related to that, why did you have only a hypothesis about nutrient effects on soil biological activity but not detritivores?

The discussion about vegetation cover and biomass in the section about herbivore effects is not well connected to the rest of the paragraph (L437-441). How does come vegetation into play? How does the finding by Borer et al. (2020) about increasing biomass with herbivore exclusion link to the soil moisture content effects?

Reviewers' comments:

Reviewer #1 (Remarks to the Author):

Review of the manuscript: "Drivers of soil biological activity across global grasslands"

In this manuscript, Siebert & Sünneberg et al. present the results of an impressive coordinated study
on the response of soil microbial and detritivore activity to nutrient addition and herbivore
reduction in 26 grasslands across 5 continents. They report that the response of the activities of both
groups to nutrient addition and herbivore reduction was mediated by the change in soil water
content, indicating that soil water content is the main driver of soil biological activity.

Overall, this is a well-executed study and a rather well-written manuscript. This research is timely,
particularly on the global drivers of detritivore activity which is largely understudied, relative to that
of soil microbial activity. I made a number of minor comments while reading the manuscript (see at
the bottom), but most importantly, I have a few general comments which I think could improve the
manuscript if integrated during the revisions. These include comments on the introduction of the
study, on the reference throughout the manuscript to literature on the biology of the organisms of
interest, on the use of climatic factors, on the lack of plant traits, and on acknowledging the
limitations of the present study.

*Response: We thank the reviewer for this encouraging feedback. We have carefully considered all*
*suggestions for improving the manuscript, integrating them into the revised version of our paper. We*
*have also taken the reviewers advice to enhance the integration of relevant literature on the*
*organisms we studied. We emphasize our use of the climatic factors and plant trait data and have*
*made sure to explicitly acknowledge the limitations of the current study. We greatly appreciate the*
*reviewer's suggestions as they have contributed to improve the overall quality and clarity of our*
*work.*

Main comments:

1. The introduction: I am struggling a bit with the introduction, which I find lacking structure and
somewhat confusing. The second paragraph starts talking about N and P depositions in the first line
and then switches to herbivory for the rest of the paragraph with no further mention of nutrient
inputs, with no knowledge gap identified, and without prior information on the importance of soil
organisms. The third paragraph introduces the importance of soil organisms at last, and the last
paragraph introduces the response of soil/soil organisms to nutrients additions. I would encourage
the authors to rethink the structure of the introduction to improve its logic. After briefly thinking
about it, I would suggest:

- Paragraph 1: largely unchanged

- Paragraph 2: introduce the importance of soil microorganisms and detritivores to soil processes,
and potential mediators of global changes. Mention the known drivers + identify the lack of
knowledge on unknown responses to herbivore pressure and nutrient addition.

- Paragraph 3: introduce the importance of herbivory and the knowns and unknowns on the
response of soil biological activity

- Paragraph 4: introduce the importance of nutrient additions (direct effect and indirect through
plant responses).

- Paragraph 5: introduce the study

*Response: We really appreciate the reviewer's suggestion and have reorganized our introduction*
*accordingly. We've also rephrased certain sections to enhance the overall flow. Since we believe that*
*the reactions of soil biota to herbivory and nutrient addition are most fitting within the paragraph*
*discussing these factors, we've structured the paragraphs by first providing a general introduction to*

*each factor and then detailing the corresponding responses. We hope that the reviewer finds this*
*approach favorable. Please see the **Introduction**. We have also rearranged our hypotheses and*
*discussion to fit this new logical flow. Please see **Lines 150-156** and **Discussion**.*

2. The biological context: Overall, throughout the manuscript (and in the introduction in particular),
much of the focus is on the broad factors of interest (nutrient input, herbivory), with little
information on the response variables: the activity of the organisms of interest. There is in fact
substantial information in the literature on the response of soil biological activity (albeit not at the
global scale of the present study). It would be good to see this information in this manuscript: Who
are these organisms? What do they do? What are detritivores? Why focus on both microbial and
detritivore activity? What do they both do respectively for soil processes? What do we know about
what drives their activity, from lab and field studies? Why might they respond differently to the
drivers mentioned, and the factors of interest? Overall, the study and its interpretation would
benefit from a more thorough presentation of the literature on the biology/ecology of soil
organisms.

*Response: We appreciate the reviewer's suggestion and like the idea to put more emphasis on both*
*organism groups. We have therefore revised parts of our introduction: "Grassland systems covering*
*approximately 40% of the world's terrestrial surface, encompass a wide variety of habitats for soil*
*organisms^{1,2}, which play key roles in delivering ecosystem functions such as nutrient cycling and*
*decomposition³⁻⁶. In this context, the key players are soil microorganisms and detritivores such as*
*earthworms, isopods, millipedes, and enchytraeids, which primarily feed on litter and organic*
*materials. Their collective efforts break down organic matter, thus supplying vital nutrients to*
*plants⁷. Soil organism activity is strongly driven by temperature, soil moisture⁸⁻¹¹, and global change*
*factors, including increased nutrient inputs and alterations in the range, abundance, and distribution*
*of aboveground herbivores. However, we lack a broad understanding of how nutrient inputs and*
*herbivory influence soil communities and ecosystem functions in grasslands. At the same time, such*
*soil organisms may be important mediators of ecosystem responses to global change^{2,12,13}. Further, a*
*lack of spatially replicated studies means that we cannot predict how plant productivity, grazing, or*
*local abiotic characteristics may mediate nutrient and herbivory effects on soil organisms¹⁴." Please*
*see **Lines 88-101**.*

3. The effect of MAP / Soil water content: I am a bit confused about the climatic factors. Was the soil
water content measured at the time of soil collection for soil microbial activity, and if so, how
representative is it of the soil water content during the period of bait lamina field incubation? Also, I
am not quite sure about the process for building the SEM, with the factors included or not after
preliminary data analyses rather than after building an a priori model based on the current
knowledge. Is the fact that MAT doesn't correlate with activity a good reason to remove it from the
SEM? Perhaps it has a negative effect on something that has a positive effect, leading to no visible
effect. In fact, I would encourage the authors to consider the aridity index (ratio of MAP to MAT) as
another climatic variable which integrate both MAP and MAT. MAP on its own is not necessarily a
good proxy of available water to soil organisms, as this depends on the level of evapotranspiration.
MAP in London (580 mm) is less than that of Paris (720 mm) which is less than that of Rome (870
92 mm), yet I wouldn't bet that Rome has a higher soil water content than London ;-)

*Response: We thank the reviewer for this valuable suggestion. We agree that MAT is an important*
*variable for soil organism activity and did incorporate it into our a priori model initially. However,*
*upon analysis, we observed that MAT showed no significant correlation with any of our response*
*variables, nor did it result in any indirect effects on soil microbial or detritivore activity. We*
*acknowledge that we did not adequately communicate this in our manuscript and have accordingly*

revised the methods section to highlight this more: “We further selected mean annual temperature
as another exogenous variable. However, the relationship between mean annual temperature (MAT)
and microbial activity was not statistically significant ($p = 0.14$), and a similar non-significant trend
was observed with detritivore activity ($p = 0.93$) (see also Figures S4c-d). Although MAT displayed a
positive correlation with plant species richness, this association did not extend to the other variables
in our SEM model (as illustrated in Figures 2 and 3). Consequently, we decided to exclude MAT from
the final model.” Please see **Lines 265-271**.

At the same time, we have also taken the reviewer's suggestion into account and included the aridity
index in our SEM. However, similar to when incorporating MAT, we observed comparable effects. The
aridity index did not show a significant association with soil microbial activity or detritivore activity. It
also did not exhibit any indirect effects, as it displayed no association with other SEM variables
except for plant species richness, which itself did not show any relationship with other variables.
Consequently, we have not included it in our final analysis.

Regardless of the climatic factors used, I was missing a discussion on why the activity of detritivores
and microbes is controlled more or less tightly by climatic variables. It may be that the soil water
content data was more aligned with the soil microbial activity measure, but it may be that
detritivores are less controlled by water content than microorganisms. Detritivores, by being able to
move, can find water in deeper soil or moist pockets in the soil, allowing them to maintain an activity
even in drier conditions. In line with that, Sagi et al. 2019 (Proc. Roy. Soc. B) found that most of the
litter decomposition in the Negev desert in the summer was carried out by a woodlice species, not
by microorganisms which didn't have enough water to grow. Interestingly, in Fig 4 it appears that
soil detritivore activity is less variable (can this be inferred from log-log relations?) than soil microbial
activity, and peaks at moderate levels of microbial activity and soil water content. This figure also
suggests that the relation between detritivore activity and soil water content is not linear.

*Response: We thank the reviewer for pointing that out. Regarding Fig. 4, here microbial and*
*detritivore activity were measured using different methods. This is why we hesitate to directly*
*compare their variability. Additionally, we believe that the peak mentioned by the reviewer might be*
*influenced by the density of data. For further insights, we examine the relationship between*
*individual activity measures in supplementary Figure 5. However, we believe that the reviewer has an*
*important point and have therefore included the reviewer's points in the discussion of our*
*manuscript: “In our study, soil water content had a significantly larger effect on soil microbial activity*
*compared to detritivore activity. This might be attributed to the fact that the soil water content data*
*were better aligned with the measure of soil microbial activity. However, it is also possible that*
*detritivores are less influenced by water content than microorganisms due to their ability to move to*
*deeper soil layers*⁷⁸*. This adaptability allows them to sustain their activity even in drier conditions. In*
*support of this, Sagi et al. 2019*⁷⁹ *discovered that the primary litter decomposition in the Negev*
*desert during summer was driven by a woodlice species, in contrast to microbes which lacked the*
*necessary water for growth.” Please see **Lines 384-392**.*

4. Plant trait data: One key parameter that would really help understand the response of soil
biological activity to these factors is plant trait data. Indeed, this resource quality is likely affected by
changes in NPK / herbivory, with important consequences for soil biological activity. This is
mentioned in the introduction but not considered later on. I am not suggesting that the authors
change their SEM to include it at this stage, but it would be good to read their thoughts on it. This
may help explain the negative relationship between plant biomass and microbial activity.

*Response: The reviewer is right that plant traits can play a significant role in influencing soil*
*biological activity, and this is an important consideration. While we did not incorporate plant trait*

*data into our current SEM, we acknowledge its potential relevance in explaining certain relationships*
*observed, such as the negative correlation between plant biomass and microbial activity. We address*
*this now in our discussion section in the revised version of our manuscript: “At the same time, we*
*observed increases in plant biomass that were associated with herbivore reduction, nutrient addition*
*and higher levels of precipitation⁸⁸. It is well-established that vegetation cover helps to maintain high*
*levels of soil biological activity, as evidenced by previous studies^{51,56,89–91}, which is consistent with our*
*own findings. However, higher plant biomass also led to declines in soil microbial activity which is, in*
*contrast to other studies reporting positive effects of higher plant biomass on soil biological activity*
*via bottom up effects of increased rhizodeposition^{19,92–94}. There are several possible explanations. On*
*one hand, enhanced plant growth could potentially result in higher transpiration rates^{95,96}, ultimately*
*leading to a reduction in soil water content over time. On the other hand, herbivore reduction also*
*led to a less diverse plant community, which could also decrease microbial respiration. It is also*
*possible that soil microbial communities have to compete with plants for nutrients, possibly leading*
*to reduced respiration rates. Follow-up studies are needed to relate environmental change-induced*
*alterations in soil microbial communities to ecosystem functions, using standardized, replicated*
*methods to increase the generality and robustness of such experiments^{97,98}.” Please see **Lines 407-***
**421.**

5. Limit acknowledgments: The methods used to assess microbial and detritivore activities are
somewhat coarse methods. Their limitations should be acknowledged and the conclusions should be
nuanced based on these limitations. Soil microbial activity was measured on soil samples that were
sampled, homogenised, sieved, shipped (how long did it take and under which conditions?), and O2
consumption measured in a centralised lab. This post-processing (disruption of aggregates), shipping
(continued microbial activity under altered conditions) and ex-situ measure of soil microbial activity
under constant temperature likely led to significant biases. This may also explain part of the lack of
correlation between microbial activity and MAT. In turn, the bait lamina, although allowing to
measure detritivore activity in situ, also comes with limitations. It was recently shown that the
disappearance of standardised substrates across multiple sites (as is the case here), as indicator of
soil biota-driven decomposition, is a poor indicator of the soil-biota driven decomposition of the
local plant litter (under realistic conditions; Joly et al. 2023, Nature Ecology and Evolution). This
question the relevance of data obtained from standardised substrates, and the validity of the
derived conclusions. This was not shown specifically for bait lamina, but it raises concerns about this
method. This is not to discredit the present study, but it would be good to see a section in the
discussion bringing up possible limitations of this study. Note that the descriptions of these methods
lack details (e.g., transport of soil for microbial activity; period during which the bait lamina were
inserted; etc.).

*Response: This is an important remark! We now acknowledge the methodological limitations of our*
*study in a separate paragraph: “Assessing soil microbial activity for a global scale study entailed*
*some constraints that could have also influenced our findings. Measuring soil microbial activity*
*involved homogenizing, sieving, and shipping soil samples to a central laboratory, which includes*
*disruption of soil aggregates, along with potential changes in microbial activity due to shipping*
*conditions. Additionally, assessing microbial activity under controlled temperature conditions,*
*differing from the natural environment, might have influenced the correlations observed with mean*
*annual temperature (MAT). Similarly, the bait lamina test, offering an on-site approach to assess*
*detritivore activity, has faced recent critique regarding its use of standardized substrates as accurate*
*indicators of local plant litter-driven decomposition¹¹⁰ even though this concern was not directly*
*tested for bait lamina strips.” Please see **Lines 437-446.***

Line 119: ‘studied’ rather than ‘studies’?

*Response: Thanks, we revised that sentence.*

Line 119: what do you mean by “macro-biota”? The associated citations rather refer to
microorganisms, a bit to soil fauna, or to arthropods in general but not specifically soil arthropods
(Lind et al. 2017; not necessarily a relevant citation here?).

*Response: We apologize for this oversight and revised wording and literature, now referring to soil*
*organisms: “At the same time, such soil organisms may be important mediators of ecosystem*
*responses to global change^{2,12,13}”. Please see **Lines 97-98.***

Line 121: It would be good to see more specific citations to the wealth of work on the function
carried out by soil animals rather than broad chapters/reviews on soil biodiversity that are not
necessarily function-focussed.

*Response: Agreed! While we kept the review paper by Bardgett and van der Putten (2014) as we*
*consider it to be quite important, we also incorporated two additional papers that directly explore*
*the role of soil microbes and invertebrates in ecosystem functioning: Gougoulas et al., 2014 and*
*Lavelle et al., 2006. Please see **Line 98.***

Line 128: This is the first time that you mention invertebrate detritivores. Can you describe what
they are/do? Some of the taxa that you refer to later in the paper are in fact rather microbial grazers
(collembola, mites) or predators (mites), rather than detritivores.

*Response: Based on the reviewer’s feedback we put more emphasis on soil organisms, talking*
*explicitly about invertebrate detritivores and also removed unsuitable citations:” Grassland systems*
*covering approximately 40% of the world’s terrestrial surface, encompass a wide variety of habitats*
*for soil organisms^{1,2}, which play key roles in delivering ecosystem functions such as nutrient cycling*
*and decomposition³⁻⁶. In this context, the key players are soil microorganisms and detritivores such*
*as earthworms, isopods, millipedes, and enchytraeids, which primarily feed on litter and organic*
*materials. Their collective efforts break down organic matter, thus supplying vital nutrients to*
*plants⁷. Soil organism activity is strongly driven by temperature, soil moisture⁸⁻¹¹, and global change*
*factors, including increased nutrient inputs and alterations in the range, abundance, and distribution*
*of aboveground herbivores. However, we lack a broad understanding of how nutrient inputs and*
*herbivory influence soil communities and ecosystem functions in grasslands. At the same time, such*
*soil organisms may be important mediators of ecosystem responses to global change^{2,12,13}. Further, a*
*lack of spatially replicated studies means that we cannot predict how plant productivity, grazing, or*
*local abiotic characteristics may mediate nutrient and herbivory effects on soil organisms¹⁴.” Please*
*see **Lines 88-101.***

Line 129: Why mention micronutrients when they are not discussed afterwards, with only mentions
of NPK?

*Response: We agree and revised our sentence accordingly: “The growth and biomass production of*
*plants depend on nutrients such as nitrogen, phosphorous, and potassium, and most grasslands are*
*limited in productivity by nutrient inputs^{38,39}.” Please see **Lines 124-126.***

Line 190: '(>50g)' What does this refer to? I don’t know what to make of it...

*Response: We apologize for this lack of clarity. Here, we refer to the fact that our fences were only*
*equipped to exclude herbivores that weigh over 50 g. We stated this more clearly in our revised*

version: "Fences excluded aboveground herbivores weighing more than 50 g." Please see **Lines 171-**
**172.**

Line 227: woodlice, no? I am not sure mites or collembola would directly feed on the substrate –
they are rather microbial grazers...

*Response: After considering the reviewer's comments, we found literature showing that it is, in fact,*
*Collembola, Enchytraeids, and earthworms that primarily contribute to the consumption of the bait.*
*We revised our manuscript accordingly and have included two references in support of this: "The bait*
*substrate is primarily consumed by soil collembolas, enchytraeids, and earthworms^{51,52}; microbial*
*activity plays a minor role in bait loss⁵³⁻⁵⁵." Please see **Lines 205-207.***

Lines starting 220: when was this done, and why the variable time of exposure (3 to 6 weeks).

*Response: The timing of the bait lamina assessment was conducted by the principal investigators*
*themselves, as per your instructions. This approach was chosen to ensure consistency in data*
*collection. Additionally, to prevent any material from becoming dislodged during transportation, we*
*followed the standard practice observed in other experiments by conducting the assessment directly*
*in the field. The variation in exposure time can be attributed to two main factors: logistical challenges*
*and differences in environmental conditions. Some study sites are visited infrequently and are*
*exceptionally difficult to access, leading to variability in the timing of assessments. Moreover, in*
*certain cases, no discernible changes were observed within a shorter timeframe, making it*
*impractical to conduct assessments at more frequent intervals. To address these variations in*
*exposure time, we have incorporated exposure time as a random effect in our statistical model. This*
*adjustment allows us to account for these factors and make appropriate corrections when analyzing*
*the data.*

*We have provided a more detailed explanation of these considerations in the revised version of our*
*manuscript: "The bait strip assessment was done by the principal investigator of each site. The bait*
*strips were inserted vertically into the soil with the uppermost hole just beneath the soil surface. A*
*steel knife was used to create a slot in the soil, before the strips were inserted. Five strips were*
*spaced 15 cm apart within each plot to account for within-plot spatial heterogeneity. After three to*
*six weeks of exposure, the bait strips were removed from the soil and directly evaluated in the field.*
*Each hole was rated as 0 (no invertebrate feeding activity), 0.5 (bait material partly consumed), or 1*
*(bait material completely consumed), based on visual inspection. Thus, soil invertebrate feeding*
*activity could range from 0 (no feeding activity) to 16 (maximum feeding activity) per strip. Mean*
*bait consumption of the five strips was calculated per plot prior to statistical analyses and expressed*
*as a percentage. Timing variations resulted from logistical challenges and environmental differences.*
*Some sites are hard to access and infrequently visited. In some cases, short intervals did not reveal*
*discernible changes." Please see **Lines 207 and 217-219.***

Lines starting 239: when was this done relative to the bait lamina tests? How were the soil samples
transported to the centralised lab?

*Response: To maintain consistency and ensure the integrity of our samples, all soil samples were sent*
*to Anita Risch in Switzerland following a standardized protocol. Subsequently, a subset of these*
*samples was forwarded to our central laboratory in Leipzig for further analysis. We took great care in*
*shipping these samples, using postal services that maintained cool temperatures and opting for fast*
*shipping methods to minimize any potential degradation during transit. These details have been*
*added to the revised manuscript to provide a clearer picture of our sample handling procedures. We*
*hope this information addresses your query and contributes to the overall clarity of our work: "Soil*
*was collected from 26 sites six weeks before peak plant biomass production (local site coordinators*

chose specific dates, as seasonality varied across different latitudes) by taking three subsamples per
plot (using a soil corer with 5 cm diameter and 12 cm depth), which were then homogenized and
sieved using a 2 mm mesh. All soil samples to Anita Risch in Switzerland following a standardized
protocol. A subset of these samples was then shipped to a centralised lab at the German Centre for
integrative Biodiversity Research in Leipzig, Germany. We ensured sample quality during transit by
using postal services with temperature control and fast shipping methods." Please see **Lines 221-228**.

Line 373: you mean herbivore *presence* was associated with lower soil moisture?

Response: Yes, thanks: "In addition, herbivore presence was associated with lower soil moisture
which may amplify effects of a drier climate. Moreover, MAP, NPK addition, and herbivore reduction
had indirect negative effects on soil microbial activity by increasing plant biomass." Please see **Lines**
**364-367**.

Line 413: "reduced"

Response: Thanks! We restructured the entire section, it now reads: "On one hand, enhanced plant
growth could potentially result in higher transpiration rates^{95,96}, ultimately leading to a reduction in
soil water content over time. On the other hand, herbivore reduction also led to a less diverse plant
community, which could also decrease microbial respiration." Please see **Lines 413-417**.

Discussion: It is somewhat confusing, as some parts discuss the response of detritivore activity and
then generalise to "biological activity", without a parallel discussion of the response of microbial
activity. Similar to the introduction, I am a bit lost in the logic of this section.

Response: We apologize for this inconsistency. Following the reviewer's feedback, we have
thoroughly revised the abstract, introduction, and discussion sections. We've made sure to explicitly
differentiate between microbial activity and detritivore activity, using the term "biological activity"
only when it pertains to both components. Additionally, we made sure to explain this in our
introduction: "In 2015, we used standardized bait (bait lamina strips) at 18 sites to assess soil
detritivore feeding activity⁴⁷, and analysed soil samples from 26 sites for soil microbial activity
(microbial respiration)⁴⁸. Throughout the manuscript, we use the term 'biological activity' to
encompass both activities." Please see **Lines 144-148**.

Line 452: Things are immobile or they are not – it is not a gradual state. Less mobile? In any case,
detritivore do not have "host plants", so these references to plant parasites/herbivores is not
relevant.

Response: We agree and have therefore removed that sentence.

Line 457: why refer to AMF (microorganisms) when this paragraph discusses the response of
detritivores?

Response: This sentence was indeed a bit misplaced here. We removed it.

Line 461: does this refer to detritivore activity, or microbial activity, or both? This should be a lot
clearer, here and throughout the discussion.

Response: Yes, as mentioned above we have addressed this in the introduction of our manuscript.

Fig. 2: Ouch! The icon in the box 'soil detritivore activity' is a centipede (one pair of leg per segment,
and modified legs at the back). Centipede are not detritivores, but predators. Please change to a
millipede for instance (two pairs of leg per segment and no modified legs at the back) – many icons
are available on line for inspiration.

*Response: We are grateful to the reviewer for bringing this embarrassing mistake to our attention. We*
*have updated our icon, thank you!*

Reviewer #2 (Remarks to the Author):

COMMSBIO-23-1077

Siebert et al. conducted an interesting global experiment to evaluate which drivers structure
detritivore and microbial activity in grasslands. The experiment included two treatments: (1)
addition of nutrients and (2) reduced grazing. In addition to these treatments, they also included
other predictors such as plant biomass and species richness and soil water content, which were in
turn analyzed as responses of other predictors with Structural Equation Models (SEMs), aiming to
disentangle direct and indirect effects on detritivore and microbial activity. They found that
detritivore activity, but not microbial activity was affected by reduced grazing. The addition of
nutrients had in both cases no direct effect, but was found to indirectly affect both detritivore
activity through regulating plant biomass and plant species richness.

Overall, I think the manuscript is well written, especially the discussion, the study is interesting and
relevant. Some aspects of the sampling and analysis were not clear to me and I suggest that the
authors improve this. I explain this in more detail below. But apart from these 'minor revisions', I
think the paper is well developed and can be published.

*Response: We thank the reviewer for the constructive feedback and these encouraging words!*

General comment:

Regarding the sampling design, there is unclarity about the treatment duration, and sampling times.
It can be understood from lines 232-233 that the invertebrate feeding activity was measured within
3 to 6 weeks, from lines 240-243 that the soil was collected 6 weeks before peak plant biomass
production and from lines 261-266 that treatments were in place for different lengths of time.
However, it is not clear from the text how long these treatments were in place and how the
sampling related to season (although collecting 6 weeks before plant biomass peak suggests
seasonal adjustment of sampling). Apparently, the treatments were sometimes in place for multiple
362 years. Please clarify sampling times, treatment lengths and the relation with season.

*Response: In our study, the timing of soil sampling was indeed coordinated with the sampling for*
*Risch et al. 2019 Nat Communication. The placement of bait lamina strips was overseen by the*
*principal investigators at each site, ensuring they could be assessed during soil sampling. Soil*
*sampling was scheduled to occur six weeks prior to the peak biomass, aiming to maintain uniform*
*conditions despite the considerable variations in latitude and seasonal patterns across our study's*
*experimental sites. We have incorporated some additional explanations in our manuscript and we*
*have added the specific exposure durations of the bait lamina strips in Table S1.*

*"Timing variations resulted from the substantial environmental differences, as in some cases, short*
*exposure intervals did not yield discernible changes (for detailed exposure time please see Table S1)."*
*Please see **Lines 216-218** and **Table S1***

*"Soil was collected from 26 sites six weeks before peak plant biomass production (local site*
*coordinators chose specific dates, as seasonality varied across different latitudes) by taking three*
*subsamples per plot (using a soil corer with 5 cm diameter and 12 cm depth), which were then*
*homogenized and sieved using a 2 mm mesh. All soil samples to Anita Risch in Switzerland following*

a standardized protocol. A subset of these samples was then shipped to a centralised lab at the
German Centre for integrative Biodiversity Research in Leipzig, Germany. We ensured sample quality
during transit by using postal services with temperature control and fast shipping methods." Please
see **Lines 221-228**.

Specific comments:

line 119 "While they have been little ..." -> "While there have been few ..."

*Response: Thanks for noticing this mistake, we have changed our wording here: ". At the same time,*
*such soil organisms may be important mediators of ecosystem responses to global change^{2,12,13}"*
*Please see **Lines 97-98**.*

Line 196-197 "... plots do not ..." -> "... plots did not ..."

*Response: We revised accordingly: "In contrast, control plots did not receive additional nutrients and*
*represent ambient soil conditions." Please see **Lines 177-178**.*

Line 211-213 Perhaps state instead of "visually estimated" that plant species richness was counted
on site. I could not find further details on this in Borer et al (2014b). Generally, the authors refer
quite often to this study for more details, but it is a very compressed Nature article. I suggest the
authors double check if these details are indeed easily found in Borer et al. (2014b).

*Response: We did as the reviewer suggested. We also revised the manuscript, trying to be more*
*careful when referring the reader to other papers for details: "Plant species richness was assessed on*
*site in a permanent 1 x 1 m quadrat located in the focal subplot in each plot." Please see **Lines 192-***

Line 232-233 Table S1 has a column with the year that the experiment started. I suggest to also add
the exact date to Table S1

*Response: Bait lamina strips were inserted by the principal investigators of each site in a way, that*
*they could be evaluated when soil sampling was taking place. The date for soil sampling was six*
*weeks before peak biomass to ensure comparable conditions despite the significant differences in*
*latitudes and seasonality across the experiments in our study. Therefore, adding the exact sampling*
*dates would not provide any new information. However, we have included the precise exposure times*
*of the bait lamina strips in Table S1 and added a note regarding the variation in exposure timing in*
*our manuscript: "Timing variations resulted from the substantial environmental differences, as in*
*some cases, short exposure intervals did not yield discernible changes (for detailed exposure time*
*please see Table S1)." Please see **Lines 216-218 and Table S1**.*

Line 240-243 While this suggests sampling was adjusted to season, it would also for microbial
activity be good to add the sampling dates to tables S1.

*Response: Soil sampling was conducted six weeks before the peak biomass to ensure comparability*
*across the various experimental sites, which encompassed a wide range of environmental conditions*
*and seasonal differences. Consequently, we opted not to specify a particular sampling date, but we*
*have included more details in the revised version of our manuscript: "Soil was collected from 26 sites*
*six weeks before peak plant biomass production (local site coordinators chose specific dates, as*
*seasonality varied across different latitudes) by taking three subsamples per plot (using a soil corer*
*with 5 cm diameter and 12 cm depth), which were then homogenized and sieved using a 2 mm mesh.*

All soil samples to Anita Risch in Switzerland following a standardized protocol. A subset of these
samples was then shipped to a centralised lab at the German Centre for integrative Biodiversity
Research in Leipzig, Germany. We ensured sample quality during transit by using postal services with
temperature control and fast shipping methods." Please see **Lines 221-228**.

Line 257-260 Are these separate models that only include NPK x Reduction or are these the models
presented as part of the SEM in Tables S3 and S4? Please clarify.

*Response: These are separate models, we clarified this in the text: "To assess the effects of*
*fertilization, reduced density of vertebrate herbivores, and their interaction on soil detritivore feeding*
*and microbial respiration across all sites without accounting for abiotic factors and plant data, we*
*employed linear mixed-effects models using the lmer function from the R-package "lme4"⁵⁷." Please*
*see **Lines 240-243**.*

Line 261-268 This section is somewhat confusing. The second sentence seems to repeat what was
stated in the first sentence ("... (2) the number of years the treatments had been in place at a
specific site to account for differences among sites ..."). But the way it is formulated sounds as if it
concerns another random effect. Also, the statement in the first sentence is confusing, as the
differences among sites was already corrected for by including site itself as a random effect. I think
the authors mean they included this to adjust for duration time of the experiment?

*Response: We agree with the reviewer and tried to make the sentence clearer, excluding repetitions.*
*Based on the next comment we also removed treatment duration from the model: "The model's*
*random intercepts were organized based on two factors: (1) block nested within site, and (2) the type*
*of O₂-microcompensation device (for soil microbial data) or field exposure duration (for detritivore*
*data), as some sites had a longer exposure time due to logistical constraints." Please see **Lines 243-***

**246**.

Finally, (lines 266-268) suggest that duration time was first included as a fixed effect and then
excluded as it was not informative. I think this is more appropriate as this would allow to model it as
a continuous variable, which would preserve the putative relationship with time, instead of turning
each year into an independent level as is done in the case of the random intercepts. If it is then
deemed not informative based on the AIC criterion or p-value, the variable can be ignored and does
not have to be included as random effects. I also wonder if rounding time length to year is optimal
and if it would not be better rounded to months or days. Please clarify this rationale behind the
decisions taken for the analysis.

*Response: We thank the reviewer for this valuable comment. By comprehensively examining the*
*model's interaction with the treatment duration, we establish that the effect doesn't intensify over*
*time. As all samples were consistently collected within the same years, we can confidently address*
*concerns about temporal correlation and consequently exclude "treatment duration" from our*
*explanatory parameters. It's important to note that our decision to remove this factor doesn't*
*diminish its significance; rather, it reflects our recognition that time alignment through simultaneous*
*sampling renders it non-influential. After reflecting on this, we have also removed "incubation time"*
*from the random effects of our model: " The model's random intercepts were organized based on*
*two factors: (1) block nested within site, and (2) the type of O₂-microcompensation device (for soil*
*microbial data) or field exposure duration (for detritivore data), as some sites had a longer exposure*
*time due to logistical constraints. We also tested a model with treatment duration years as a fixed*
*effect, but found that treatment duration had no significant impact on soil microbial and detritivore*

activity, and consequently excluded "treatment duration" from our explanatory parameters." Please
see **Lines 243-249**.

line 315-316 It could be interesting to also calculate this correlation in the based on the residuals of
the detritivore activity and microbial activity models, because as the color scale in Figure 4 also
suggest this correlation is probably mainly driven by water content and it might be interesting to see
how the responses are correlated after controlling for the effects of the studied variables.

*Response: We show in our SEMs that microbial and detritivore activity are strongly associated with*
*soil water content. In Fig.4, we aim to demonstrate a connection between soil water content,*
*microbes, and detritivores. By considering residuals, we might inadvertently introduce causality.*

Figures Generally, the figures are very pixelated and in some cases difficult to read. It is
recommended to present vector figures.

*Response: Thanks! We did as the reviewer advised. Please see all **Figure 1,2,3 and 4**.*

Fig. 1 Are the presented data points raw observations or partial residuals?

*Response: They are raw observations, we clarified this in the text: "Fig. 1: a) Global map of all*
*participating sites in the study. Red dot = data on soil microbial and detritivore activity (n = 18 sites);*
*blue dot = data on soil microbial activity only (n = 26 sites). b,c) The effects of NPK fertilization and*
*herbivore reduction on soil detritivore activity (log-scaled) and soil microbial activity (log-scaled).*
*Points are raw observations; error bars indicate 95% confidence intervals." Please see caption of*
**Figure 1**.

Reviewer #3 (Remarks to the Author):

The manuscript shows the results of the global, well-designed experimental set up NutNet where
effects on nutrient addition and herbivore reductions have been monitored. The submitted study
disentangles the effects and mechanisms of both treatments on soil biological activity finding that
despite no direct treatment effects, treatment effects were mediated via their influence on soil
water content and plant biomass. I find this study well-written and analysed showing the processes
and context-dependency behind global change drivers (eutrophication, grazing) on soil microbial
activity i.e., the base for ecosystem functioning.

*Response: We thank the reviewer for the positive feedback on our study and manuscript! We have*
*implemented the reviewer's suggestion into the revised version.*

I have only minor comments on the presentation of the introduction, statistical methods, the results,
and parts of the discussion.

Introduction:

L106: Here you mention changes in herbivore densities. You look at a reduction treatment, how does
this fit together with replacing wild by domestic herbivores. Reading this one could think that
herbivore pressure globally is not reduced then and one should look rather on effects of increased
herbivore pressure. How could you put this statement of shifts in herbivores (wild vs domestic) into
the context of your treatment?

*Response: We recognize the importance of discussing domestic livestock animals due to their*
*significant role in modern farming practices. Nonetheless, our primary focus remains on investigating*

*the impacts stemming from the absence of both wild and domestic herbivores. This absence could*
*potentially yield either negative or positive effects, which we aim to explore. We rephrased the*
*sentence and hope to enhance clarity: “At the same time, large native herbivore densities may be*
*reduced via hunting or land conversion, and in many cases, they are replaced by large numbers of*
*domestic livestock^{21,22}. Soil communities, processes, and structure are strongly affected by wild and*
*domestic herbivores, with important consequences for soil biological activity and ecosystem*
*multifunctionality^{14,23–25}.” Please see **Lines 108-112**.*

L129: I was struggling in this section on nutrient stoichiometry to see it’s “added value”. All the
information is relevant but partly I felt it repeats things said in the sections before. Was it to mention
context-dependent effects of grazing, grazing mechanisms? Maybe just a better topic sentence is
missing?

*Response: We agree and have revised the entire section and the structure of the discussion to avoid*
*repetitions and provide more clarity: “ Predictions suggest that the disruption of the nitrogen cycle*
*could cause nitrogen (N) deposition to double in the future^{34,35}. The same applies to phosphorus (P)*
*inputs, which have globally increased compared to preindustrial levels^{36,37}. The growth and biomass*
*production of plants depend on nutrients such as nitrogen, phosphorous, and potassium, and most*
*grasslands are limited in productivity by nutrient inputs^{38,39}. Nitrogen inputs may increase the activity*
*of soil organisms by increasing the amount and quality of plant material that enters the soil*
*system^{10,20}, but have also been shown to reduce detritivore activity⁴⁰. The same applies to soil*
*microbes, as long-term nitrogen inputs have been shown to have negative effects⁴¹.” Please see*
**Lines 122-129**.

Statistical methods:

When introducing the SEM approach, the selection of environmental variables was not very clear. I
was missing some more overview about selection criteria for environmental variables, why e.g., soil
organic carbon was not included which is known in other studies as an important driver. You
mention this in your discussion, too (L409). Probably this can be simply solved by restructuring this
section a bit, mentioning the selection of environmental covariates and reason for their selection in
the beginning of the section. Did you base the selection on previous findings or on correlation of
those parameters with the response variables? Information like this I was missing. This would give
the reader simply a better overview of the logic of you SEM.

*Response: Thanks for this suggestion, we have implemented it in the revised version of our*
*manuscript: “We used structural equation modeling (SEM) to disentangle direct and indirect pathway*
*effects by which fertilization and herbivore reduction affected the activity of soil organisms. In*
*determining the environmental variables for our SEM approach, we considered factors that could*
*offer meaningful insights into the dynamics of soil microbial and detritivore activity. These variables*
*were selected a priori based on their established influence on soil microbial and detritivore activity,*
*as well as their potential to mediate treatment effects. Our choices were guided by existing literature*
*in the field. Given that plant community composition and biomass are strong predictors of soil*
*microbial and detritivore activity and are thus likely to mediate the treatment effects, we included*
*plant species richness and total plant biomass in the SEM (Fig. 2 and 3). As soil organisms are highly*
*dependent on soil moisture⁵⁹, we included soil water content as a key abiotic driver. We also chose to*
*include mean annual precipitation (MAP) as another exogenous variable, as it was significantly*
*correlated with the two soil activity variables (Figs. 2 and S4a-b) and should have long-lasting effects*
*on soil conditions that are also relevant for our snapshot assessments. We further selected mean*
*annual temperature as another exogenous variable. However, the relationship between mean annual*
*temperature (MAT) and microbial activity was not statistically significant ($p = 0.14$), and a similar*

*non-significant trend was observed with detritivore activity ($p = 0.93$) (see also Figures S4c-d).*
*Although MAT displayed a positive correlation with plant species richness, this association did not*
*extend to the other variables in our SEM model (as illustrated in Figures 2 and 3). Consequently, we*
*decided to exclude MAT from the final model.” Please see **Lines 252-271.***

Finally, in the abstract it was mentioned that soil biological activity was measured on 28 and
detritivore activity only on 16 grasslands. Please, mention the different sample sizes in the methods
section again.

*Response: That’s important, we agree and have implemented it: “Soil invertebrate feeding activity*
*was assessed at 16 sites using the bait lamina test (Terra Protecta GmbH, Berlin, Germany), which is*
*commonly used as a rapid ecosystem function assessment method^{11,47}.” Please see **Lines 202-204.***

*And: “Soil was collected from 26 sites six weeks before peak plant biomass production (local site*
*coordinators chose specific dates, as seasonality varied across different latitudes) by taking three*
*subsamples per plot (using a soil corer with 5 cm diameter and 12 cm depth), which were then*
*homogenized and sieved using a 2 mm mesh.” Please see **Lines 221-224.***

Results:

The figure captions of the SEM do not introduce the content in the three different panels. This could
be solved with one sentence telling what is shown in panel a, b and c.

*Response: We revised accordingly: “b) Direct, indirect, and net effect of MAP on soil detritivore*
*activity, and c) direct, indirect, and net effect herbivore reduction on soil detritivore activity.” Please*
*see **Figure 2.***

*And: b) Direct, indirect, and net effect of MAP on soil microbial activity, and c) direct, indirect, and*
*net effect herbivore reduction on soil microbial activity, and d) direct, indirect, and net effect of NPK*
*fertilization on soil microbial activity.” Please see **Figure 3.***

L325 How can it be that the LMM shows no effect of herbivory, but the SEM does show a direct
effect?

*Response: The reviewer may have missed it, but the herbivory actually had a (albeit marginally)*
*significant effect on soil detritivore activity. Please see **Figure 1.***

Discussion:

You start the discussion with the topic of soil moisture (L381-394), which is the most prominent in
your results, I agree. Yet, this topic occurs within the other sections again, as it’s the cause for
indirect effects of the main factors - fertilization and herbivory. To avoid this redundancy, I suggest
to - after the opening paragraph – start straight away with the sections about hypothesis 1 and 2 and
include the explanation on soil moisture within them.

*Response: We like this idea a lot and have implemented it! Please see **Discussion paragraphs 2 and***
***3.***

In relation to hypothesis 2 (L418...) you mention that it was partially confirmed through the direct
herbivory effects on detritivore activity. As the hypothesis includes also soil biological activity, I was

missing at least one sentence mentioning that this effect was not found – or only in a direct form.
Otherwise, I was stuck wondering about why the hypothesis was only partially confirmed.

*Response: Thanks for pointing that out, we have revised this part accordingly: “In line with our*
*hypothesis (1), we found a consistent overall positive effect of herbivore reduction on soil detritivore*
*activity across all sites. In addition, through our SEM analysis, we observed that herbivore reduction*
*enhanced soil detritivore activity directly and also indirectly via an increase in soil water content.*
*However, while herbivore reduction had no direct effect on soil microbial activity, it also indirectly*
*increased microbial activity via increases in soil water content.” Please see **Lines 372-377.***

Related to that, why did you have only a hypothesis about nutrient effects on soil biological activity
but not detritivores?

*Response: We apologize for any confusion regarding these terms. We have clarified our distinction*
*between soil microbial activity and soil detritivore activity in specific sections. When addressing them*
*both we refer to soil biological activity, which we now state in the revised version of our introduction:*
*“In 2015, we used standardized bait (bait lamina strips) at 18 sites to assess soil detritivore feeding*
*activity⁴⁷, and analysed soil samples from 26 sites for soil microbial activity (microbial respiration)⁴⁸.*
*Throughout the manuscript, we use the term 'biological activity' to encompass both activities.”*
*Please see **Lines 144-147.***

The discussion about vegetation cover and biomass in the section about herbivore effects is not well
connected to the rest of the paragraph (L437-441). How does come vegetation into play? How does
the finding by Borer et al. (2020) about increasing biomass with herbivore exclusion link to the soil
moisture content effects?

*Response: Agreed! We restructured and revised parts of the discussion to improve the clarity of this*
*section: “At the same time, we observed increases in plant biomass that were associated with*
*herbivore reduction, nutrient addition and higher levels of precipitation⁸⁸. It is well-established that*
*vegetation cover helps to maintain high levels of soil biological activity, as evidenced by previous*
*studies^{51,56,89–91}, which is consistent with our own findings. However, higher plant biomass also led to*
*declines in soil microbial activity which is, in contrast to other studies reporting positive effects of*
*higher plant biomass on soil biological activity via bottom up effects of increased rhizodeposition^{19,92–}*
*⁹⁴. There are several possible explanations. On one hand, enhanced plant growth could potentially*
*result in higher transpiration rates^{95,96}, ultimately leading to a reduction in soil water content over*
*time. On the other hand, herbivore reduction also led to a less diverse plant community, which could*
*also decrease microbial respiration. It is also possible that soil microbial communities have to*
*compete with plants for nutrients, possibly leading to reduced respiration rates. Follow-up studies*
*are needed to relate environmental change-induced alterations in soil microbial communities to*
*ecosystem functions, using standardized, replicated methods to increase the generality and*
*robustness of such experiments^{97,98}.” Please see **Lines 407-421** and the **Discussion** in general.*

REVIEWERS' COMMENTS:

Reviewer #1 (Remarks to the Author):

I am fully satisfied by the revisions made by the authors on their manuscript.

Reviewer #2 (Remarks to the Author):

The authors have revised their manuscript accurately. They have clearly considered all the issues I raised in the first revision round carefully and have provided appropriate solutions. I have no further points that need to be addressed. I congratulate the authors with a very nice and interesting study which I believe is ready for publication.

Reviewer #3 (Remarks to the Author):

The only thing I came across is that in Fig 1 they do not explain the significance levels (*) and NS in the caption.